# Translation and Psychometric Properties of the Strategies Used by People to Promote a Health Instrument for the Assessment of Self-Care Self-Efficacy among Patients Undergoing Hemodialysis in Vietnam

**DOI:** 10.3390/healthcare11111644

**Published:** 2023-06-04

**Authors:** Thi Thuy Nga Nguyen, Shu-Yuan Liang, Chieh-Yu Liu, Huu Dung Nguyen

**Affiliations:** 1Faculty of Nursing and Midwifery, Hanoi Medical University, No 1, Ton That Tung Street, Dong Da District, Hanoi 116177, Vietnam; 2School of Nursing, National Taipei University of Nursing and Health Sciences, 365 Ming Te Road, Peitou, Taipei 112, Taiwan; 3College of Health Technology, National Taipei University of Nursing and Health Sciences, 365 Ming Te Road, Peitou, Taipei 112, Taiwan; 4Nephro-Urology-Dialysis Center, Bach Mai Hospital, No 78, Giai Phong Street, Dong Da District, Hanoi 116177, Vietnam

**Keywords:** end-stage renal disease, hemodialysis, reliability, self-care, self-efficacy, validity

## Abstract

Self-care and self-efficacy play an important role in predicting quality of life among patients undergoing hemodialysis, but there currently is a lack of an instrument in the Vietnamese language for assessing self-care and self-efficacy. This limits the ability of researchers to explore and determine the confidence patients have in their ability to perform relevant self-care activities. The purpose of this investigation was to assess the validity and reliability of the Strategies Used by People to Promote Health questionnaire-Vietnamese version. This cross-sectional study involved translation, validation, and cultural adaptation of the questionnaire into Vietnamese and a trial with 127 patients undergoing hemodialysis in Bach Mai Hospital (Hanoi, Vietnam). The questionnaire was translated by bilingual translators and validated by three experts. Internal consistency and confirmatory factor analysis were applied. This questionnaire demonstrated good content validity and a Cronbach’s alpha of 0.95 for the total scale. Confirmatory factor analysis of the three-factor model showed moderate model fit (comparative fit index = 0.84, Tucker–Lewis coefficient = 0.82, root mean square error of approximation = 0.09). Overall, this questionnaire exhibited acceptable validity and reliability for measuring self-care and self-efficacy among patients undergoing hemodialysis.

## 1. Introduction

The management of end-stage renal disease (ESRD) requires kidney transplantation or dialysis [1]. In Vietnam, the incidence of ESRD continues to rise by approximately 8000 cases annually [2]. According to the Dean of Bach Mai Hospital’s Artificial Kidney Department, 1.3% of new ESRD cases require long-term treatment in the form of hemodialysis [2]. Consequently, patients may face a number of challenges in applying strategies for self-care; for instance, around 40% of kidney dialysis patients never or rarely undertake physical exercise [3]. The study conducted by Wong et al. [4] showed that patients face learning challenges during the training process. Patients may feel sleepy during dialysis, which makes it difficult for them to focus during training. Patients may also have difficulty obtaining peer support for communication and gaining relevant self-care experience. Patients’ opinions regarding their self-care ability play an important role in how they face treatment regimens. Increasing self-efficacy is commonly considered the most effective approach to changing individual behavior [5].

Self-efficacy is a key factor affecting patients’ healthcare behaviors [6,7]. Within the framework of social cognitive theory [8], as applied to a patient’s health self-care, self-efficacy refers to a patient’s beliefs or judgments about his/her ability to perform certain health self-care behaviors [8]. Self-efficacy plays a key self-regulatory role in the patient’s efforts toward self-set health goals [9]. Based on the self-efficacy theory, the patient’s judgment of their own ability to affect health care will affect his/her emotions, motivations, behaviors, and experienced environmental pressure [8]. Self-efficacy also plays a pivotal role in the process of health self-care. When a patient encounters a difficult situation, the patient’s behavioral choice to face or avoid the difficulty, how much effort to expend, and the degree of persistence of their will are related to the patient’s own self-efficacy. Thus, self-efficacy plays an important role in the process of a patient’s implementation of a specific health self-care behavior. Self-efficacy influences event-processing outcomes in individuals facing difficult situations [8].

Self-care self-efficacy is task-specific and adapted from the self-efficacy concept in the context of self-care [10]. For patients with ESRD undergoing hemodialysis, self-care self-efficacy is defined as the patient’s confidence in his or her ability to perform relevant self-care activities [11]. Individuals who are confident in their abilities approach difficult tasks as challenges to overcome rather than as threats to avoid. In contrast, those who lack confidence in their abilities avoid challenging tasks, quit quickly, and focus on failure [8]. Moreover, self-care self-efficacy is an important factor for predicting quality of life [12].

The Strategies Used by People to Promote Health (SUPPH) instrument has been used to evaluate the confidence in self-care activities of patients. The term self-care self-efficacy was utilized for the first time in a 1996 study by Lev and Owen [13]. The initial instrument, comprising 29 items, was developed in English and tested in patients receiving treatment for cancer and ESRD [13]. The initial model [13] included four factors (coping, stress reduction, decision-making, and enjoying life) that showed good psychometric properties. Subsequently, this scale was merged into three factors and used with regard to the treatment of various types of cancer and ESRD, as well as SARS, stroke, and HIV-AIDS [14]. The Chinese version of this instrument has demonstrated excellent validity and reliability [15]. The SUPPH has also been translated and validated in Farsi, exhibiting acceptable validity and reliability [16]. However, there is currently a lack of such instruments in Vietnamese. This limits the ability of researchers to explore and assess the self-efficacy of patients undergoing hemodialysis in Vietnam.

Questionnaires or scales are important tools in clinical practice or nursing research. Questionnaires can be an efficient, reliable, and low-cost way to gather patients’ subjective information [17,18,19]. When a questionnaire is to be applied in another language, cultural and environmental factors must be considered in addition to language issues. The process of translation and cultural application of a questionnaire must thus be standardized to ensure that the translated version is consistent with the original. The psychometric properties of the new version must then be evaluated to ensure that the new version preserves the characteristics, reliability, and validity of the original questionnaire [19,20,21].

There is a need for early identification and intervention with regard to individuals lacking self-care and self-efficacy. There is a requirement for a translated and validated version of SUPPH in Vietnam (SUPPH-V). The objective was to provide an instrument for measuring self-care and self-efficacy among patients undergoing hemodialysis. The purpose of this research was to translate the SUPPH into Vietnamese and validate the psychometric properties of the SUPPH-V by assessing content validity, internal consistency, item-total correlation, and construct validity.

## 2. Methods

### 2.1. Study Design, Sample, and Procedure

This was a cross-sectional study based on convenience sampling. Patients undergoing hemodialysis for at least 3 months at the Artificial Kidney Department of Bach Mai Hospital (Hanoi, Vietnam) were recruited. All patients were literate and agreed to participate in this study. Patients with cognitive impairment, frail patients, and those dependent on caregivers were excluded from the study. The study was conducted from October 2020 to June 2021. Approximately 20–30 min were required to complete the questionnaire. When the patient had completed the questionnaire, the investigator checked whether there were any items that had been omitted and, if so, asked the patient to complete the missing items.

### 2.2. Ethical Considerations

This study was approved by the Institutional Review Board of Hanoi University of Public Health, Hanoi, Vietnam (approval number: 020-400/2020/DD-YTCC). The researcher explained the purpose and method of this study to the patients. The patients provided written informed consent for their participation in this study. Thereafter, the researcher invited patients to complete the self-administered questionnaire or interviewed them to assist in this process. Personal information was coded in the questionnaire to protect patient privacy. Patients who were unwilling to continue the survey or were thought unfit for further evaluation owing to poor physical condition could withdraw from the study. In such cases, data collection by the researcher was discontinued.

### 2.3. Instrument

#### 2.3.1. Strategies Used by People to Promote Health Questionnaire (SUPPH)

This original SUPPH is used to translate to SUPPH-V. Lev and Owen [13] developed the SUPPH instrument to evaluate the self-care and self-efficacy of patients with ESRD and cancer. This scale was reliable, with a Cronbach’s alpha for internal consistency reliability of 0.93. A correlation between the SUPPH and quality of life (r = 0.34, *p* < 0.01) was observed, indicating the validity of this instrument [13]. Construct validity was reported through factor analysis that identified four factors (coping, stress reduction, making decisions, and enjoying life) that explained 81% of the variance [13]. Subsequently, the creators of the SUPPH revised the confirmatory measurement model by combining two highly related factors into a new category (i.e., “coping” and “enjoying life” became “positive attitude”) [22]. They demonstrated that there was no appreciable difference between the three- and four-factor models [22]. The three-factor model includes the following subscales: stress reduction (10 items: 1–10), decision making (three items: 11–13), and positive attitude (16 items: 14–29). Each item is rated using a 5-point scale ranging from 1 (very little confidence) to 5 (quite a lot of confidence). The score is computed by calculating the mean of the responses to all items within each scale. Higher scores reflect better self-care and self-efficacy.

#### 2.3.2. Strategies Used by People to Promote Health Questionnaire-Vietnamese Version (SUPPH-V)

This study got permission to translate the SUPPH to SUPPH-V. This study used a translation and back-translation process recommended by Younan et al. [23] (who followed the World Health Organization’s guidelines) to produce the SUPPH-V. This process involved four steps: forward translation; back-translation; pre-testing; and final version [23]. Initially, the English version of the questionnaire was translated into Vietnamese by a bilingual translator. Then, the researcher revised the translation through comparison with the original English version. Subsequently, a different bilingual translator translated the Vietnamese version back into English without having seen the original English version of the questionnaire. Finally, a native speaker compared the two English versions to ensure that the meaning was identical. Based on this process, a final SUPPH-V was produced for validation by an expert panel.

### 2.4. Data Analysis

We used SPSS version 22 (IBM, Armonk, NY, USA) for the analysis of descriptive statistics and internal consistency. AMOS version 22 was used to perform a confirmatory factor analysis (CFA) to assess the construct validity of the SUPPH-V [24].

#### 2.4.1. Internal Consistency

Internal consistency refers to the extent to which all items in a scale measure the different aspects of a single attribute [25]. Cronbach’s alpha is the most commonly used measure for assessing the internal consistency of continuous variables [26]. Cronbach’s alpha ranges from 0 to 1, with ≥0.7 denoting sufficient reliability [27]. A value ≥0.8 is considered to indicate good reliability [28].

#### 2.4.2. Content Validity

We used validity assessment by experts to estimate if the items of this translated scale could reflect the self-care and self-efficacy of Vietnamese hemodialysis patients. It has been suggested that the minimum and maximum acceptable numbers of experts for assessing content validity are 2 and 10, respectively, but most guidelines suggest a minimum of 6 experts [29]. Each panel member rated content validity, independently scoring each item for relevance using a scale ranging from 1 to 4 (1 = not relevant; 2 = somewhat relevant; 3 = quite relevant; 4 = very relevant) [30]. The panel members rated 29 items of the translated SUPPH. There are two forms for calculating the content validity index (CVI), namely CVI for the item (I-CVI) and CVI for scale (S-CVI). The I-CVI refers to the proportion of content experts giving an item a relevance rating of 3 or 4. With three to five experts on the panel, the judged standard for I-CVIs should be 1.00 [31]. The researcher calculated the I-CVI by summing items that received this rating from all experts and dividing the value by the number of experts. Furthermore, there are two methods for calculating S-CVI/Ave (scale-level CVI based on the average method) [29]. The first approach is the average of the I-CVI scores for all items on the scale or the average of proportion relevance judged by all experts. The second is the average relevance rating by an individual expert. The present study calculated S-CVI/Ave as an average of I-CVI scores. The judged standard for the S-CVI/Ave should be ≥0.90 [31]. The experts were asked to provide comments for improving the relevance of items to the targeted domain.

#### 2.4.3. Construct Validity

CFA was used to assess construct validity to test the hypothesis of a relationship between the observed variable and its underlying structure. It is primarily used to check whether a particular indicator or topic is categorized along the dimensions expected based on theoretical considerations or empirical research [32].

We used four major indicators: (1) absolute fit index: root mean square error of approximation (RMSEA); (2) an absolute fit measure: goodness-of-fit index, the ratio of chi-square score to degrees of freedom (ꭓ²/df); (3) an incremental fit index: comparative fit index (CFI); and (4) the Tucker–Lewis index (TLI). RMSEA values of 0.05–0.08, 0.08–0.10, and >0.1 indicate “good fit”, “moderate fit”, and “no fit”, respectively, and the target standard is “good fit”. TLI values close to 1 indicate a “very good fit” [32].

## 3. Results

A total of 127 participants were recruited. The mean age of the participants was 51.4 years (range: 21–84 years), and 52% of the participants were female. The average dialysis vintage was 97.8 months. Values (mean and standard deviation) for each item in the three subscales are shown in Table 1.

### 3.1. Internal Consistency

The findings showed that the internal consistency of the SUPPH-V total scale was high, with a Cronbach’s alpha of 0.95. The Cronbach’s alphas for the three subscales (stress reduction, decision-making, and positive attitude) were 0.92, 0.71, and 0.92, respectively.

In item analysis, all corrected item-total correlation coefficients for the stress reduction subscale (items 1–10) were >0.4. For the decision-making subscale, most of the corrected item-total correlation coefficients were >0.6, except for that of item 13 (0.29). After excluding item 13, the recalculated Cronbach’s alpha was increased to 0.86. For the positive attitude subscale, most of the corrected item-total correlation coefficients were >0.4, except for that of item 20. Following the exclusion of item 20, Cronbach’s alpha was not increased substantially. A correlation coefficient ≥0.4 denotes a moderate to high correlation. The results are shown in Table 2.

### 3.2. Content Validity

The expert panel that reviewed the SUPPH-V and assessed the content validity of the instrument consisted of an assistant professor with a background in nutrition and health sciences, a Ph.D. holder in nursing who has experience working and conducting research in the community, and a Master’s degree holder in nursing with experience working and conducting research at the Department of the Nephro-Urology of Bach Mai Hospital. The three experts gave a rating of 3 or 4 to all items. Thus, the I-CVI and S-CVI/Ave values of all items were 1.00.

### 3.3. Construct Validity

The original model did not exhibit a good fit. After modifying some items with high modification indices covariances (i.e., 14 and 15, 16 and 19, 21 and 22), the fit of the second model was acceptable. CFA of the three-factor model revealed a moderate fit (CFI = 0.84, TLI = 0.82, RMSEA = 0.09). The indices obtained from the CFA for the second model are presented in Table 3.

## 4. Discussion

The purpose of this study was to create a Vietnamese version of SUPPH through translation and back-translation and to investigate its reliability and validity for assessing the self-care and self-efficacy of patients undergoing hemodialysis. Developing a new measurement questionnaire is a time-consuming process; the translation and cultural applicability assessment of existing instruments facilitate their use by clinical experts or researchers worldwide and can also serve in the comparison of research results from different countries [20,21]. In order to ensure that the characteristics of the translated questionnaire are similar to those of the original, the present study followed the suggestions of researchers such as Younan et al. [23] during the translation standardization process. The guidelines made the process of translation possible to ensure a high level of cross-cultural and cross-language applicability. After the back-translation was completed, a teacher whose native language was English judged that the back-translation and the original text were similar in meaning, indicating that the translation carried out here did not alter the meaning of the original scale items. The high Cronbach’s alpha value across all items (0.95) showed that SUPPH-V was sufficiently reliable. However, this value also indicates that the scale questions might be risking repetition in scale items or are excessively long and should be shortened [26,33]. Indeed, reliability depends on the number of items included in a scale; hence, a higher number of items is associated with better reliability. Cronbach’s alpha for the decision-making subscale was 0.71 and thus lower than that of the other scales, probably because this subscale included only three items. Importantly, the exclusion of item 13, “Deciding for myself whether or not to have treatment”, increased Cronbach’s alpha of the decision-making scale. Several previous studies have also demonstrated that SUPPH is characterized by acceptable reliability [15,16,34,35,36,37].

In this study, we calculated S-CVI/Ave as an average of the I-CVI. For this purpose, the sum of I-CVI scores was divided by the number of items. The I-CVI and S-CVI/Ave of all items were 1.00, indicating that this questionnaire has good content validity. Therefore, this questionnaire can be utilized for the assessment of self-care and self-efficacy among patients undergoing hemodialysis. However, the expert panel suggested that this questionnaire might be misunderstood. The items show the level of confidence of patients while they perform the actions but do not indicate the frequency of those actions. Therefore, a detailed explanation of this distinction to patients by the data collectors is necessary before conducting the survey.

The factorial structure of the SUPPH-V was assessed via a three-factor model (i.e., stress reduction, decision-making, and positive attitude). In the first model, several items had high modification indices covariances, such as items 14 and 15, 16 and 19, and 21 and 22. There may be perceived overlaps in the meaning of these pairs of items that lead to the error covariance. Obviously, items 14 and 15 were about life attitude, items 16 and 19 were about stress issues, and items 21 and 22 were about beliefs themselves. Following modification, the second model had an acceptable CFI score (0.84) and TLI score (0.82) and a moderate RMSEA (0.09). However, the goodness-of-fit index (0.70) showed that this study was limited by its small sample size. Overall, the results show that the three-factor model had acceptable construct validity and that these items can be combined in future studies.

Yuan et al. [15] used a three-factor model for the Chinese version of SUPPH. According to CFA, this model exhibited a better fit than the two- and four-factor models. In that study, the three-factor model showed stronger model fit indices (CFI = 0.94, TLI = 0.94, RMSEA = 0.05) than those obtained in the present study. These discrepancies can be attributed to differences in the populations examined in these studies. The study conducted by Yuan et al. included oncology patients and a larger sample size (n = 764). This comparison supports Bandura’s theory of self-efficacy. According to Bandura, self-efficacy is an individual’s belief that they can successfully perform a specific behavior [38]. The thoughts and feelings of individuals influence their behavior, which is determined by their confidence in their own self-efficacy [39]. An increase in self-efficacy is the most effective approach to changing individual behavior [5]. This core capability for belief in self-efficacy is the basis of human motivation, accomplishments, and emotional well-being [8]. Patients undergoing hemodialysis are subjected to long-term treatment. Hence, their self-care depends strongly on maintaining a positive attitude or hope, reducing stress, and making appropriate decisions.

Overall, SUPPH-V constitutes a valid and reliable instrument that can be used to assess self-care and self-efficacy among patients undergoing hemodialysis in Vietnam. Using this questionnaire, healthcare professionals may detect low self-care and self-efficacy in patients and work to enhance it. Self-efficacy theory suggests that the four principal elements of self-efficacy in persons are performance, vicarious experience, verbal persuasion, and self-appraisal [8]. Healthcare professionals can incorporate these four elements into the design of interventions to enhance patients’ self-efficacy and use the questionnaire developed here to assess patients’ improvements in self-care and -efficacy before and after an intervention. Healthcare professionals can develop appropriate interventions that could enhance patient confidence in their self-care activities as well as improve their quality of life [40].

## 5. Study Limitations

Some limitations of this study should be acknowledged. Firstly, patients were recruited from only a single hospital by convenience sampling, and we recommend that future studies include patients from more hospitals or institutions. Secondly, this study had a relatively small sample size for confirmatory factor analysis. Bentler and Chou [41] suggested that 5–10 participants should be involved per questionnaire item. To obtain a stable factor construction, Tabachnick and Fidell [42] suggested enrolling at least 300 participants. In fact, the basis for sample size considerations is still a matter of debate [43]. Research revealed a range of sample size requirements (i.e., from 30 to 460 cases) based on different research parameters, such as the number of indicators and factors [43]. Finally, this study was performed over a short period of time based on a cross-sectional design without a “test-retest” structure in the reliability assessment. Test-retest reliability refers to the same subject, using the same measurement tool, and testing twice at different times and is used to reflect the stability of the measuring tool [44]. However, this method is susceptible to memory or practice effects, and the interval between two tests should be appropriate [45]. Investigators are usually required to specify reasonable intervals between repeated measurements [45]. This study initially verified the reliability and validity of the SUPPH-V scale. Further research is therefore required to confirm the present findings.

## 6. Conclusions

The total scale and all three subscales of SUPPH-V, including a total of 29 items, showed good internal consistency. This instrument exhibited good content validity and acceptable construct validity. SUPPH-V may be useful to nurses for assessing the views of patients regarding self-care behaviors and may provide healthcare professionals with preliminary data for future research.

## Figures and Tables

**Table 1 healthcare-11-01644-t001:** Descriptive statistics for each item (n = 127).

No.	Items	Mean	SD
Subscale 1: Stress reduction			
1	Excluding upsetting thoughts from my consciousness	2.97	1.17
2	Using relaxation techniques to decrease my anxiety	2.65	1.04
3	Finding ways to alleviate my stress	2.65	1.12
4	Using a specific technique to manage my stress	2.71	1.11
5	Doing things that helped me to cope with previous emotional difficulties	2.60	1.06
6	Practicing stress reduction techniques even when I’m feeling sick	2.63	1.01
7	Managing to keep anxiety about illness from becoming overwhelming	3.06	1.01
8	Thinking of myself as better off than people who became ill when they were younger than I am now	3.42	1.05
9	Focusing on something not associated with my illness as a way to decrease my anxiety	2.87	0.95
10	Believing that using a technique to manage treatment stress will actually work	3.07	0.96
Subscale 2: Decision making			
11	Choosing the treatment that seems right for me among treatment alternatives recommended by my physician	2.68	1.13
12	Making my own decision regarding treatment alternatives	2.69	1.25
13	Deciding for myself whether or not to have treatment	3.40	0.91
Subscale 3: Positive attitude			
14	Experiencing life’s pleasures since I became ill	2.49	1.11
15	Doing special things for myself to make life better	2.35	1.11
16	Convincing myself that I can manage the treatment stress	3.07	1.06
17	Helping other people going through illness and treatment	3.39	1.12
18	Convincing myself that the treatment is not so bad	3.54	0.92
19	Keeping my stress within healthy limits	3.01	0.94
20	Appreciating what is really important in life	3.61	0.69
21	Believing I can find strength within myself for healing	3.13	1.01
22	Convincing myself I’ll be O.K.	3.32	0.91
23	Finding a way to help me get through this period	2.80	0.87
24	Believing that I really have a positive attitude about my state of health	3.62	0.88
25	Doing things that helped me to cope with previous physical difficulties	2.72	0.91
26	Doing things to control my fatigue	2.68	0.88
27	Finding ways to help myself feel better if I am feeling blue	2.80	0.87
28	Managing the side effects of treatment so that I can do things I enjoy doing	2.82	0.94
29	Dealing with the frustration of illness and treatment	3.22	0.89

SD, standard deviation.

**Table 2 healthcare-11-01644-t002:** Cronbach’s alpha and corrected item-total correlation for SUPPH-V (n = 127).

Items	Corrected Item-Total Correlation	Cronbach’s Alpha after Item Exclusion	Cronbach’s Alpha
Subscale 1: Stress reduction			
1	0.61	0.91	
2	0.70	0.91	
3	0.83	0.90	0.92
4	0.80	0.90	
5	0.76	0.90	
6	0.75	0.91	
7	0.75	0.91	
8	0.47	0.92	
9	0.72	0.91	
10	0.55	0.92	
Subscale 2: Decision making			
11	0.64	0.49	
12	0.73	0.33	0.71
13	0.29	0.86	
Subscale 3: Positive attitude			
14	0.56	0.92	
15	0.46	0.92	
16	0.66	0.92	
17	0.67	0.92	
18	0.61	0.92	
19	0.75	0.91	0.92
20	0.35	0.92	
21	0.58	0.92	
22	0.59	0.92	
23	0.74	0.91	
24	0.76	0.91	
25	0.74	0.91	
26	0.71	0.92	
27	0.74	0.91	
28	0.61	0.92	
29	0.55	0.92	
Total scale			0.95

**Table 3 healthcare-11-01644-t003:** Confirmatory factor analysis (CFA) model.

Model	Index
ꭓ²/df	2.05
*p*-value	<0.001
GFI	0.70
TLI	0.82
CFI	0.84
RMSEA	0.09

ꭓ²/df: ratio of chi-square value to degrees of freedom, GFI: goodness-of-fit index, TLI: Tucker–Lewis index, CFI: comparative fit index, RMSEA: root mean square error of approximation.

## Data Availability

Data presented in this study are available from the first author upon reasonable request.

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
