# Peer review of "Translation and Psychometric Properties of the Strategies Used by People to Promote a Health Instrument for the Assessment of Self-Care Self-Efficacy among Patients Undergoing Hemodialysis in Vietnam"

_healthcare, 2023, doi:10.3390/healthcare11111644_

Round 1
Reviewer 1 Report
The paper identifies the validity and reliability of the Strategies Used by People to Promote Health questionnaire-Vietnamese version (SUPPH-V).
The topic of the manuscript is within the scope of the Journal and could be relatively valuable to the scientific audience.
This study bridging the specific gaps in the field. The present investigation of Health questionnaire-Vietnamese version (SUPPH-V) showed that cultural and environmental factors were considered in addition to language issues. The process of translation and cultural application of a questionnaire has been standardized to ensure that the translated version is consistent with the original. The psychometric properties of the new version have been evaluated to ensure that the new version preserves the characteristics, reliability, and validity of the original questionnaire. As a novelty, this study examined patients undergoing hemodialysis.
The quality of the research design is acceptable. The title of the article is accurate.
Abstract reflects the work done and the conclusions drawn.
Some clarifications are however needed:
INTRODUCTION
Research hypothesis is missing. Authors should provide justification for the research hypothesis.
METHOD
Could authors provide some justification of the sample size being enough? The small sample size can lead to extreme sensitivity of estimation.
Please address the following issues:
Have the assumptions for CFA been tested and met? For instance, multivariate normality, a sufficient sample size (n >200), the correct a priori model specification, and data must come from a random sample.
RESULTS
Findings are presented in a correct way. However, study was performed without a “test-retest” structure in the reliability assessment.
I think that explanation must be given about the possible disruption of experimental work throughout these challenging times in the DISCUSSION (Limitations) section.
DISCUSSION
I suppose that the limitation of the study about missing test-retest in the reliability assessment must be more clearly defined.
I think that the future perspectives should be more precisely described.
TO SUM UP I think the author(s) need to make the recommended corrections.
Author Response
Response to Reviewer 1 Comments
Comments and Suggestions for Authors
The paper identifies the validity and reliability of the Strategies Used by People to Promote Health questionnaire-Vietnamese version (SUPPH-V).
The topic of the manuscript is within the scope of the Journal and could be relatively valuable to the scientific audience.
This study bridging the specific gaps in the field. The present investigation of Health questionnaire-Vietnamese version (SUPPH-V) showed that cultural and environmental factors were considered in addition to language issues. The process of translation and cultural application of a questionnaire has been standardized to ensure that the translated version is consistent with the original. The psychometric properties of the new version have been evaluated to ensure that the new version preserves the characteristics, reliability, and validity of the original questionnaire. As a novelty, this study examined patients undergoing hemodialysis.
The quality of the research design is acceptable. The title of the article is accurate.
Abstract reflects the work done and the conclusions drawn.
Some clarifications are however needed:
INTRODUCTION
Research hypothesis is missing. Authors should provide justification for the research hypothesis.
Response: Research hypotheses are generally used to describe the relationship between variables. The current study tends to describe the research results of a single variable (SUPPH-V), so we prefer to provide research purpose. We add the research purpose at the end of Introduction.
METHOD
Could authors provide some justification of the sample size being enough? The small sample size can lead to extreme sensitivity of estimation.
Response: We add “In fact, the basis for sample size considerations is still a matter of debate [43]. Research revealed a range of sample size requirements (i.e., from 30 to 460 cases) based on different research parameters such as number of indicators and factors [43].”
Please address the following issues:
Have the assumptions for CFA been tested and met? For instance, multivariate normality, a sufficient sample size (n >200), the correct a priori model specification, and data must come from a random sample.
Response: Random sampling in clinical human care research has its difficulties, for example, difficulty in obtaining lists of target populations for register numbering. Although convenience sampling was used in this study, univariate normality was met. In fact, the basis for sample size considerations for CFA is still a matter of debate [43]. Research revealed a range of sample size requirements (i.e., from 30 to 460 cases) based on different research parameters such as number of indicators and factors [43]. We add this issue in Study Limitation.
RESULTS
Findings are presented in a correct way. However, study was performed without a “test-retest” structure in the reliability assessment.
I think that explanation must be given about the possible disruption of experimental work throughout these challenging times in the DISCUSSION (Limitations) section.
Response: Yes, this study did not test the test-retest reliability. We add some new explanation about test-retest reliability. We had put it in the section of Study Limitations and suggest further research is required to confirm the present findings (see p.8).
DISCUSSION
I suppose that the limitation of the study about missing test-retest in the reliability assessment must be more clearly defined.
Response: We add more explanation about test-retest reliability in Limitation.
I think that the future perspectives should be more precisely described.
Response: We do the corrections for the Conclusion.
TO SUM UP I think the author(s) need to make the recommended corrections.
Response: We do the corrections according to the suggestions of the reviewers.

Reviewer 2 Report
Authors in this study intended to translate and cross-culturally adapt the Vietnamese version of the SUPPH questionnaire. They are able to determine that the content validity and internal consistency are adequate.
However, I think there are several issues they should work on before considering this study for publication.
- The title could include "translation" or "cross-cultural adaptation" as this is a relevant methodological process, and could use a generic "psychometric properties" instead of mentioning the validity and reliability.
- Abstract: In the Abstract please do not include acronyms, this is discouraged when reporting studies. Instead, use the full name of the instrument
- Introduction. lines 33-34. What about peritoneal dialysis? Use the term "dialysis" instead of hemodialysis
- Line 39. Please change to "The study conducted by Wong"
- Line 48-49. You should cite Albert Bandura after this sentence
- Line 70. Could you reference here the initial model?
- Methods. Line 95. You don't have to state the sample size in Methods, only in Results section is appropriate
- Line 104. I think this part could be included within the last one, it doesn't need to have its own subsection.
- line 106-107. Use "researcher" instead, more appropriate term
- line 114. The translation section should be written before this section, and you should call this section "Outcomes"
- line 115. Use the full name of the instrument
- line 131. Due to license property, it is mandatory ask and have the permission of the original developer of the questionnaire to conduct a translated version of it. Please add a statement in case you've contacted with the original developers, Lev and Owen. If not, this is a major concern.
- line 172. I'm missing other validity assessments, such as convergent validity
- Methods. Also, I'm missing another common psychometric property assessment method testing the test-retest reliability of the instrument. This assessments could've been furtherly developed
- Results. line 184. Please refer to them as "participants", every time you state "patients", please change the term
- line 189. You should be consistent and name this section "internal consistency"
- Discussion. lines 226-227. You could discuss here why did you choose this guidelines instead of more commonly used ones, such as the ones developed by Beaton et al. or the COSMIN guidelines
Overall, I see merit behind the work, but there are several elements that must be improved to pursue publication in Healthcare.
Author Response
Response to Reviewer 2 Comments
Comments and Suggestions for Authors
Authors in this study intended to translate and cross-culturally adapt the Vietnamese version of the SUPPH questionnaire. They are able to determine that the content validity and internal consistency are adequate.
However, I think there are several issues they should work on before considering this study for publication.
- The title could include "translation" or "cross-cultural adaptation" as this is a relevant methodological process, and could use a generic "psychometric properties" instead of mentioning the validity and reliability.
Response: We change the title to “Translation and Psychometric Properties of the Strategies Used by People to Promote Health Instrument for the Assessment of Self-Care Self-Efficacy Among Patients Undergoing Hemodialysis in Vietnam”.
- Abstract: In the Abstract please do not include acronyms, this is discouraged when reporting studies. Instead, use the full name of the instrument
Response: We remove this acronym and use the full name of the instrument.
- Introduction. lines 33-34. What about peritoneal dialysis? Use the term "dialysis" instead of hemodialysis
Response: We use "dialysis" instead of hemodialysis.
- Line 39. Please change to "The study conducted by Wong"
Response: We change to "The study conducted by Wong".
- Line 48-49. You should cite Albert Bandura after this sentence
Response: We cite Albert Bandura after this sentence.
- Line 70. Could you reference here the initial model?
Response: We reference here (Line 70.) the initial model.
- Methods. Line 95. You don't have to state the sample size in Methods, only in Results section is appropriate
Response: We delete the statement of sample size in Methods and add the sample size in Results section.
- Line 104. I think this part could be included within the last one, it doesn't need to have its own subsection.
Response: We prefer to keep it here to connect the Research Procedure and to show the subsection of Ethical Considerations.
- line 106-107. Use "researcher" instead, more appropriate term
Response: We use "researcher" to instead “investigator”.
- line 114. The translation section should be written before this section, and you should call this section "Outcomes"
Response: We change the subtitle “Translation and Back-Translation” to subtitle “Strategies Used by People to Promote Health questionnaire-Vietnamese version” to correspond to the “2.3 Instrument”.
- line 115. Use the full name of the instrument
Response: We use the full name of the instrument to instead SUPPH.
- line 131. Due to license property, it is mandatory ask and have the permission of the original developer of the questionnaire to conduct a translated version of it. Please add a statement in case you've contacted with the original developers, Lev and Owen. If not, this is a major concern.
Response: We add a statement “This study got the permission to translate SUPPH to SUPPH-V”. We had got the permission to translate SUPPH. We can provide proof of this authorization if necessary.
- line 172. I'm missing other validity assessments, such as convergent validity
Response: Yes, this study did not mention convergent validity. This study initially verified the reliability and validity of the SUPPH-V scale. We suggest further research is therefore required to confirm the present findings (see p.8).
- Methods. Also, I'm missing another common psychometric property assessment method testing the test-retest reliability of the instrument. This assessments could've been furtherly developed
Response: Yes, this study did not test the test-retest reliability. We add some new explanation about test-retest reliability. We had put it in the section of Study Limitations and suggest further research is required to confirm the present findings (see p.8).
- Results. line 184. Please refer to them as "participants", every time you state "patients", please change the term
Response: We change “patients” to "participants".
- line 189. You should be consistent and name this section "internal consistency"
Response: We change “Reliability” to “Internal Consistency”.
- Discussion. lines 226-227. You could discuss here why did you choose this guidelines instead of more commonly used ones, such as the ones developed by Beaton et al. or the COSMIN guidelines
Response: We address this issue in discussion.
Overall, I see merit behind the work, but there are several elements that must be improved to pursue publication in Healthcare.
Response: We improve our works according to the suggestions of reviewers.
